# Developing a Physical Activity Intervention Strategy for Pregnant Women in Buffalo City Municipality, South Africa: A Study Protocol

**DOI:** 10.3390/ijerph17186694

**Published:** 2020-09-14

**Authors:** Uchenna Benedine Okafor, Daniel Ter Goon

**Affiliations:** 1Department of Nursing Sciences, University of Fort Hare, 50 Church Street, East London 5201, South Africa; 2Department of Public Health, University of Fort Hare, 5 Oxford Street, East London 5201, South Africa; dgoon@ufh.ac.za

**Keywords:** physical activity, intervention strategy, pregnant women, Buffalo City Municipality, South Africa

## Abstract

Despite global awareness about the importance and health benefits of physical activity (PA) during pregnancy, several studies have reported a low prevalence of PA participation among pregnant women in both developed and developing countries. This means that most pregnant women do not meet the current PA recommended guideline of 150 min of moderate intensity PA per week. The global call to prioritise PA participation levels in the general population necessitates evaluating the factors affecting PA practice. Seemingly, pregnant women mostly from low-to-middle income countries like South Africa are often predisposed to adverse pregnancy outcomes, possibly because of limited access to, and knowledge of, improved pregnancy and health outcomes as a result of PA participation. Physical activity has been sparsely studied among pregnant South African women, and specifically, there is no known study that assesses the PA levels, patterns, beliefs, sources of information, perceived benefits, barriers, attitudes of pregnant women concerning PA and exercise participation; nor one that explores the perspectives of healthcare providers regarding prenatal PA in the Eastern Cape Province. In addition, no PA intervention strategy exists to promote PA participation in the region. This study, in attempting to fill these gaps in knowledge, adopts two phases. In Phase I, a concurrent mixed-method (quantitative and qualitative) approach assesses the following factors related to PA participation in pregnant women: participation levels, beliefs, attitudes, perceived benefits, barriers to uptake and sources of information. It further ascertains if healthcare professionals are sufficiently informed about PA and if they are advising pregnant women about the need for PA participation during pregnancy. Data will be collected through a structured questionnaire, interviews and focus group discussions. Information on socio-demographic and maternal characteristics will be obtained, and the Pregnancy Physical Activity Questionnaire (PPAQ) will assess PA during pregnancy. A sample size of 384 pregnant women is the required minimum sample for an infinite population at a confidence level of 95%, a precision level of ± 5% and at a prevalence of PA or exercise during pregnancy of 50% (*p* < 0.05); however, a sample size larger than the minimum number necessary will be recruited to account for possible attrition and to protect against possible data loss. Data will be analysed using a multiple logistic regression to determine the factors that predict sedentary or moderate PA levels and chi-squared analysis to determine the associations between the PA levels of the participants and socio-demographic and clinical variables. The study will assess the data collected on the above-mentioned variables and draw conclusions based on patterns and themes that emerge during analysis. Phase II of the study focuses on strategy development and validation to facilitate the promotion of PA during pregnancy. The developed strategy will be validated through the application of the Delphi technique and the administration of a checklist to selected key stakeholders through organised workshops. Understanding the level and correlates of PA participation among this special population is fundamental to designing intervention strategies to enhance their understanding of, and participation in, PA and exercise. Furthermore, this study’s findings will inform facility-based healthcare providers about the need to integrate health education on PA and pregnancy into antenatal and postnatal care visits in the setting.

## 1. Introduction

Pregnancy confers a unique period in a woman’s lifetime and, therefore, maternity health is a global health priority [1]. Instead of just focusing on minimising the direct causes of maternal morbidity, the focus has now shifted to correcting the modifiable health risk factors (e.g., diet and physical activity (PA)) [2]. Physical inactivity is reported to be the fourth leading risk factor for non-communicable diseases (NCDs) and it contributes to the global burden of disease [3]. Considering the escalating rate of NCDs attributable to physical inactivity, the World Health Organization (WHO) launched a global action plan to reduce physical inactivity by at least 10% by 2025 and 15% by 2030 [3,4,5]. Globally, most women do not meet the 150 min per week PA recommendation by specialised bodies and institutions. Previous studies have reported the prevalence of prenatal inactiveness with variations across countries as follows: Serbia (27.2%) [6], USA (25%) [7], China (45.15%) [8], Ethiopia (21.9%, 8.4%) [9,10], Nigeria (10.2%, 13.6%) [11,12] and Norway (14.6%) [13].

Several studies have reported the benefits of PA participation during pregnancy. These include reduced risk of excessive gestational weight gain [14,15,16,17] and of developing conditions such as gestational diabetes mellitus (GDM) [15,18,19,20], pre-eclampsia [15,20,21,22], lower levels of maternal pregnancy triglycerides and a higher concentration of high-density lipoprotein cholesterol (HDL-c), [23] and neonatal cord blood HDL-c levels [24]. Lower rates of preterm birth [25,26,27], miscarriage [28], reduced length of labour, lowered risk of caesarean delivery and complications [21,25,27,29,30], reduced risk of macrosomia [28,31] and reduced risk of low birth weight [25] are also counted as benefits. These also extend to the mother’s increased cardiac output, ventilation and energy expenditure [32], improved sleep [33,34,35], and reduced fatigue, stress, anxiety and depression [28,36,37,38,39]. Other benefits include reduced lower back pain and lumbopelvic pain [40,41,42,43], PA helping to maintain—or increase—cardiovascular endurance, muscle strength, resistance, agility, coordination and equilibrium [44,45], and improved overall well-being [21]. A recent study has shown that higher levels of PA by pregnant women are associated with improved breastfeeding outcomes [46].

Notwithstanding the substantial evidence on the benefits of PA during pregnancy, studies have reported a considerable decline in PA among pregnant women in both developed [47,48,49,50] and developing countries [6,9,10,51,52,53,54,55], with varying degrees of participation shaped by context-specific paradigms.

In South Africa, a previous study reported that only 17% of pregnant women were physically active [56]. Another study on the perceived role and the influencers of PA among pregnant women from low socio-economic status communities reported that 44% of pregnant women were physically inactive during pregnancy; however, of the 56% who reported doing some PA, 44% reported participating in light PA and 12% in moderate PA [57]. This suggests low prenatal PA, and few pregnant women in South Africa fulfilled the authoritative guidelines of PA during pregnancy, perhaps due to varying contextual reasons. Understanding the factors influencing the behaviour of pregnant women toward PA during pregnancy is important for the development of effective targeted health promotion strategies.

Despite the benefits of PA during pregnancy and after childbirth, pregnant women rarely meet the exercise recommendations during pregnancy. Worryingly, less than 50% of women adhere to the prenatal PA guidelines set by various bodies to promote a healthy pregnancy [58]. The intrapersonal factors limiting the participation of pregnant women in PA include low levels of maternal awareness and education on the benefits of PA during pregnancy [2,57], pregnancy symptoms and discomfort [2,57,59,60], multiparity [61], lack of strength or fatigue [57,59,62,63], lack of time [57,59,62,64], lack of motivation [57,62,63,65], lack of self-confidence [57,66] and lack of safety or fear [57,67,68]. Interpersonal or social factors affecting PA during pregnancy are cultural and religious beliefs [69,70], lack of social support [71,72], having child care responsibilities [59,71,73] and work and family responsibilities [49,59,62,63].

Studies have also reported environmental barriers to PA during pregnancy, citing lack of access to facilities and resources [62,74,75] and bad weather conditions [71,76]. Understanding the factors affecting PA during pregnancy is crucial in the provision of quality antenatal and obstetric healthcare services to this specific population. However, this requires an empirical understanding of the context-specific factors affecting PA involvement during pregnancy. This kind of information is lacking in the Eastern Cape Province of South Africa.

Compared to other regions in the world, there are only a few published studies on PA among pregnant women in Africa [9,10,11,56,57,66,68,77]. On this continent, pregnancy is often considered a time of confinement and many women’s activities and exercise levels decrease over the course of the pregnancy due to tiredness or being unwell [56]. It is estimated that 27.5% of the African population do not meet the recommended guidelines for PA [54]. Similarly, many pregnant women in Africa do not live up to the recommendations for PA, as their participation in PA is low [9,10,11].

The terms “physical activity” and “exercise” are sometimes used interchangeably; however, these two concepts do not necessarily mean the same thing. Physical activity entails “all muscle-induced bodily movements leading to an increase in energy expenditure above ∼1.0/1.5 MET (metabolic equivalent of task; 1 MET = 1 kcal (4184 kJ) • kg^−1^ • h^−1^)” [78,79,80]. Exercise, on the other hand, is a subset of PA; it is planned, structured and repetitive activity [78,79,80]. Physical activity or exercise can be used to improve or maintain general health and to achieve therapeutic health outcomes. In this context, physical intervention can be viewed as a general term encompassing both PA and exercise [79]. Given the scarcity of research on prenatal PA in the context of the Eastern Cape Province, our study’s aim is to assess PA and its correlates during pregnancy and, further, to develop an intervention strategy for PA and exercise participation during pregnancy among women in Buffalo City Municipality (BCM), Eastern Cape, South Africa.

## 2. Research Questions

The study seeks answers to the following questions:What are the patterns and correlates of PA of women during pregnancy?What are the beliefs and sources of information about PA and exercise of women during pregnancy?What are the perceived benefits of PA and exercise participation of women during pregnancy?What are the barriers to PA and exercise participation of women during pregnancy?What knowledge and attitudes do pregnant women have about PA and exercise?What beliefs, knowledge and practices do healthcare providers have toward PA and exercise participation of women during pregnancy? Do women speak to their healthcare providers about PA and get recommendations? Do healthcare providers prescribe PA restrictions or choose to restrict activity during prenatal sessions?What context-specific intervention strategy would be relevant to enhance the understanding and promotion of PA and exercise, and to mitigate the barriers associated with PA and exercise participation during pregnancy among women in Buffalo City Municipality?How valid would the developed physical activity intervention strategy be in the promotion of physical activity and exercise among pregnant women in Buffalo City Municipality?

## 3. Theoretical Framework: Theory of Planned Behaviour

Factors influencing PA participation by pregnant women are complex and multidimensional. It is important to understand the social, cognitive and behavioural factors predicting and describing PA participation, in order to inform an intervention strategy targeted specifically at pregnant women [2]. As such, the theory of planned behaviour (TPB) is a relevant approach for understanding, explaining and predicting PA behaviours [81,82], including during pregnancy [83]. In designing an intervention strategy for the promotion of PA among pregnant women, it is imperative to understand the multidimensional determinants of PA participation, and the TPB offers a useful framework within which to operate. Given the scarcity of research on the determinants and beliefs of South African women regarding PA during pregnancy, and particularly in the context of the Eastern Cape Province, the application of the TPB provides a formative theory on which to base the development of effective PA interventions aimed at pregnant women.

This study is premised on the TPB, which assumes that, to predict a pregnant woman’s intention to participate in PA during pregnancy, it is important to establish her attitude to and belief in PA. However, the practicality of participating in PA among pregnant women is questionable because there are various demographic, physical, environmental, psycho-social and cultural challenges faced by pregnant women in the Eastern Cape. Therefore, the study seeks to examine the perceived factors and predictors and to develop an intervention strategy. Hence, the aim of this study is to develop an intervention strategy for PA and exercise participation during pregnancy among women in Buffalo City Municipality, South Africa. At the end of the intervention, the expected outcomes include an improvement in PA levels, along with an improvement in lifestyle behaviours, with good clinical outcomes and more effective weight control during pregnancy.

## 4. Problem Statement

Despite the clear guidelines and recommendations set by various bodies and institutions, PA participation remains a challenge, not only to the general population in South Africa, but specifically to the special population group of pregnant women. Additionally, the literature is clear on the benefits of PA to pregnancy outcomes; yet, anecdotal evidence indicates pregnant women in Buffalo City Municipality seldom participate in PA or exercise. Whatever the reason for their non-participation in PA or exercise during pregnancy may be, it has not been investigated. Possible reasons may be ignorance, false or unscientific beliefs about PA and pregnancy, lack of awareness concerning PA guidelines during pregnancy, as well as lack of support and encouragement from health professionals.

There are studies to guide interventions and the promotion of PA among pregnant women in different countries around the world [11,51,52,54,65,66,67,68,73,76,84,85,86,87,88,89], with varying degrees of PA participation and with underpinning reasons for and against PA participation during pregnancy. However, PA among South African women is sparsely studied; and the few studies [2,53,56,57,77] failed to utilise a large sample, with a heterogeneous population, and were conducted in only two provinces and/or urban geographical settings. In addition, the aim of their empirical investigations was not to design any intervention strategy for PA participation by pregnant women. Of particular note is that there is no known study that assesses the PA levels, beliefs, sources of information, perceived benefits, barriers, attitudes of pregnant women concerning PA and exercise participation in the Eastern Cape Province. The present study is conceptualised to assess and examine the multidimensional factors influencing PA participation of women during pregnancy in Buffalo City Municipality in the Eastern Cape Province. Based on the information derived from the findings, a physical activity intervention strategy will be developed to encourage and promote physical activity and exercise during pregnancy in the context of the Eastern Cape Province. This is relevant and important in keeping with the ethos of primary healthcare paradigm–health promotion rather than health cure. In addition, such information will potentially add to the body of knowledge in this area of maternal health and further guide PA interventions tailored to pregnant women in South Africa.

## 5. Methods and Designs

Aim and objectives: The overall aim of this study is to assess PA and its correlates during pregnancy and further to develop an intervention strategy for PA and exercise participation during pregnancy among women. The following objectives are defined:

### 5.1. Phase I (Empirical Investigations)

To assess the physical activity patterns of pregnant women and to verify what characteristics (age, residential status, race, marital status, education, employment status, religion, social support, smoking, alcohol use, parity, mode of pregnancy delivery, PA advice and pregravid weight status) are related to physical activity.To examine the beliefs and sources of information about PA and exercise of women during pregnancy.To examine the perceived benefits of PA and exercise of women during pregnancy.To examine the barriers to PA and exercise of women during pregnancy.To assess the knowledge and attitudes about PA and exercise of women during pregnancy.To explore the beliefs, knowledge and practices of healthcare providers toward PA and exercise participation during pregnancy.

### 5.2. Phase II (Intervention Strategy)

To design and develop intervention strategies to enhance the understanding and promotion of PA and exercise, particularly during pregnancy, and to mitigate the barriers associated with PA and exercise participation of women during pregnancy.

### 5.3. Phase III (Validation of Intervention Strategy)

To validate the intervention strategies to promote participation of PA and exercise participation of women during pregnancy.

The first phase focuses on empirical mixed data collection from the participants, and the data will be analysed to inform the second phase that concerns the development and validation of strategies that would promote PA participation during pregnancy. The research approach is summarised in Figure 1.

## 6. Study Setting

The study will be conducted in Buffalo City Municipality, situated on the East Coast of the Eastern Cape Province, South Africa. The Eastern Cape Province was created in 1994, and include areas from the Xhosa homelands of the Transkei and Ciskei, as well as part of the Cape Province. The Eastern Cape Province is one of the poorest provinces in South Africa [90]. The Eastern Cape Province is made up of two metropolitan municipalities, namely Buffalo City and Nelson Mandela Bay Metropolitan Municipalities and six districts, i.e., Alfred Nzo, Amathole, Chris Hani, Joe Gqabi, OR Tambo and Sarah Baartman [90]. The study will be conducted at the antenatal health units of the 12 selected primary healthcare centres in Buffalo City Municipality District, in the Eastern Cape Province, South Africa. The Buffalo City Metropolitan Health District is made up of two provincial hospitals (Frere and Cecilia Makhiwane), two district hospitals (Bhisho and Grey), five community health centres; 72 primary health clinics as well as 12 mobile health services [91]. The basic antenatal healthcare services are free of charge, and offered exclusively from the community health centres and primary health clinics. Pregnant women attend these health facilities regardless of ethnicity, geographical residence and socio-economic status.

## 7. Participant Recruitment and Study Population

### 7.1. Pregnant Women

The target population will include all pregnant women aged 18 years or older attending antenatal clinics at the primary healthcare centres in the Buffalo City Municipality. A two-stage sampling technique will be applied to recruit pregnant mothers in the study. First, 12 antenatal primary health clinics will be selected by simple random sampling; second, pregnant women who fulfil the inclusion criteria will be conveniently selected because of cost and easy accessibility since the study will be conducted at the antenatal clinics and the sampling frame was unknown. Pregnant women will need to meet the following inclusion criteria: they must (i) be aged 18 years or older; (ii) be receiving antenatal care; (iii) have a single pregnancy (not multiples); (iv) be in their 2nd or 3rd trimester; and (v) have the ability to read or understand the IsiXhosa, Afrikaans or English languages. Participants will be excluded from the study if found to have disabilities or pre-existing health conditions that would prevent or limit the effect of PA at the time of recruitment. One or more contraindications for PA [92] are as follows:“Persistent excessive shortness of breathSevere chest painRegular and painful uterine contractionsVaginal bleedingPersistent loss of fluid from the vagina indicating rupture of the membranesPersistent dizziness or faintness that does not resolve on rest”

### 7.2. Healthcare Providers

In addition, healthcare providers offering antenatal healthcare services to pregnant women in their antenatal healthcare facilities will be purposively sampled to ensure variation in age, work place experience and geographical location. The inclusion criteria will be health providers who have practiced for more than a year and who are able to read and write in English.

## 8. Data Collection Instruments

### 8.1. Quantitative Data Collection

#### 8.1.1. Demographic, Obstetric and Lifestyle Information

A structured and pretested questionnaire will be developed and interviewer-administered to all study participants. The questionnaire will be prepared in English. A face-to-face interview will be employed to gather information on age, residence, ethnicity, marital status, level of education, employment status, religion, family support and behavioural and lifestyle characteristics, such as current exposure to alcohol and smoking. In addition, information on parity, mode of pregnancy delivery and pregravid body mass index will be obtained from the available antenatal records. Other information will include antepartum haemorrhage, chronic illness, physical activity advice from health provider and physical activity engaged in before pregnancy. The Institute of Medicine (IOM) has recommended BMI cut-off values that will be used to classify underweight (29.0 kg/m^2^), normal weight (19.9–26.0 kg/m^2^), overweight (26.1–29.0 kg·m^2^) and obese (>29.0 kg/m^2^) [93].

Other secondary outcome measures will include a structured questionnaire to obtain information on beliefs, attitudes, perceived benefits, barriers to uptake and sources of information on PA and exercise during pregnancy.

#### 8.1.2. Outcome Measure: Physical Activity

The primary outcome measure is the active and inactive prenatal PA participation. Physical activity will be measured by the completion of the Pregnancy Physical Activity Questionnaire (PPAQ) [94], which has been validated for various countries [11,94,95] and which is reliable for the measurement of PA in pregnant women [96]. The PPAQ will solicit participants’ information on the time spent on participation in 32 activities in different categories including household and care-giving (13 activities), occupational (five activities), sports and exercise (eight activities), transportation (three activities) and inactivity (three activities). The type, intensity, duration and frequency of PA will be recorded as hours and minutes per day. The PPAQ will be interviewer-administered to the participants at the selected health facilities during an antenatal visit and will take approximately 20 to 25 minutes. The metabolic equivalent of task (MET) of each activity will be categorised as sedentary (<1.5 METs), low or light (from 1.5 to ≤3.0 METs), moderate (3.0–6.0 METs) and vigorous intensity (>6.0 METs) [94].

#### 8.1.3. Sample Size

A sample size of 384 pregnant women is the required minimum sample for an infinite population at a confidence level of 95%, a precision level of ±5% and at a prevalence of PA or exercise during pregnancy of 50% (*p* < 0.05) [97]. The calculation for required sample size is as follows: *p* = 0.5 and hence *q* = 1 − 0.5 = 0.5; *e* = 0.05; *z* = 1.96.

Thus:$$n_{0}{}_{=}\;\frac{\left(1.96\right){}^{2}\;\left(0.5\right)\left(0.5\right)}{\left(0.05\right){}^{2}}\;384.16=384$$

However, a sample size larger than the minimum number necessary will be recruited to account for possible attrition and to protect against possible data loss.

### 8.2. Qualitative Data Collection

#### 8.2.1. Pregnant Women

A qualitative descriptive approach is deemed relevant, enabling the understanding the factors that determine behaviour in the context of PA participation during pregnancy. Pregnant women attending antenatal healthcare visits in the selected health clinics in the setting will be conveniently selected to participate in the qualitative interviews. Semi-structured individual interviews and four focus group discussions (FGDs) of eight participants each will be conducted with pregnant women in English. Interviews and FGDS will be conducted with the pregnant women until saturation (no new emerging facts or information). Using interview guides, pregnant women will be interviewed about their beliefs, attitudes, perceived benefits, barriers and sources of information on PA and exercise during pregnancy at the selected health facilities during antenatal visits. Interviews will be conducted at the health clinics, and interviews will last between 45 and 60 min each. The interviews for pregnant women will be conducted for three months. Interviews will be conducted at the antenatal health clinics in a quiet room provided by the health facility manager. Permission will be obtained from the pregnant women to audio-record the interviews and the recordings were transcribed verbatim afterwards. The interview guide will not be provided to the participants; nevertheless, participants will be provided the chance to freely express their views or opinions on beliefs, attitudes, perceived benefits, barriers and sources of information on PA and exercise during pregnancy, beyond the items in the interview guide. To maintain accuracy and validity, the transcriptions will be cross-checked with the audio-recorded interviews, and likewise, participants will be provided with the interviews’ transcripts and emergent themes for comments and confirmation regarding accuracy and veracity of the interviews. In addition, the participants will be invited to confirm the transcripts and themes.

#### 8.2.2. Healthcare Providers

Similarly, semi-structured face-to-face interviews will be conducted with the health providers offering antenatal healthcare services to pregnant women attending the selected antenatal health clinics until saturation. Participants will be provided with an information sheet, an informed consent form and a demographic questionnaire (gender, ethnicity, work place, years of practice, knowledge about current international guidelines and recommendations for PA). Participants who consent to participate will be requested to return their signed consent form and completed demographic questionnaire to the principal investigator (U.B.O.). The participants will be contacted by phone concerning the date, time and venue for their respective interviews. The interview guide will be based on extensive literature on knowledge, attitudes and practices of health providers toward PA participation during pregnancy. The interview guide will be tested in a pilot interview with a practicing community midwife, after which the semi-structured interview guide will be revised, if necessary. The interviews will be conducted across two months, and each interview will last between 35 and 45 min. Interviews will be conducted in English. Interviews will be conducted at the health facility (participants’ places of work) in a quiet designated place provided by the health facility manager. All interviews will be audio-recorded, with permission of the participants, and the recordings will be transcribed verbatim afterwards. The specific topics of the interview guide will focus on the health providers’ perspective on knowledge, attitudes and practices regarding counselling pregnant women on PA during pregnancy in relation to:Knowledge on prenatal PA, particularly the American College of Obstetricians and Gynaecologists (ACOG) guidelines and recommendationCounselling on general aspects of PA (benefits, contraindications, types of PA or exercises, intensity, duration, types of antenatal exercises or activities)Barriers and facilitators for PAPregnant women’s concerns about PAPrescription of activity restriction

The interview guide will not be provided to the participants. However, every participant will be granted the opportunity to freely express their views or opinions on prenatal PA, beyond the items in the interview guide. The transcriptions will be cross-checked with the audio-recorded interviews for accuracy. To ascertain the validity of the data, some of the participants will be provided with the interviews’ transcripts and emergent themes for their comments and confirmation of their accuracy and veracity. The participants will be invited to endorse the transcripts and themes, should they wish to do so.

### 8.3. Data Analysis

Quantitative and qualitative data will be collected and analysed separately in order to produce two sets of findings. The findings will be combined and compared according to a triangulation technique suggested by other authors [98,99].

#### 8.3.1. Quantitative

Two aspects of statistical analysis will be applied to the collected data. First, descriptive statistics using frequency distribution, percentages, mean and standard deviation (SD) will be used to summarise the data. Applying the Centers for Disease Control and Prevention (CDC) recommendations, women will be categorised as “inactive” (reporting 0–149 min of exercise per week) and “active” (reported 150 min or more of PA) [100].

Second, all bivariate and multivariate analyses will be performed on two categories of participants, classified as either inactive or active, to determine the factors affecting PA behaviour during pregnancy. Associations between the PA levels and socio-demographic, lifestyle and obstetrics characteristics will be determined using chi-squared analysis. The chi-squared test will also be used to test associations between the knowledge and the attitudes of women toward PA and exercise and the participants’ characteristics. The odds ratio (OR) and corresponding confidence interval (CI) of 95% will be calculated. A multiple logistic regression, with an automatic variable selection procedure will be used to determine the factors that predict PA levels. The significance level will be set at *p* = 0.05. The Statistical Package for Social Sciences (SPSS) (Version 24.0, IBM SPSS, Chicago, IL, USA) will be used to perform all statistical analyses.

#### 8.3.2. Qualitative

Focus group discussions and interviews will be analysed using the content analysis method, according to the steps as described by Tesch [101]. All interviews will be audio-recorded, with permission of the participants, and the recordings will be transcribed verbatim afterwards. The audio-recorded interviews will be cross-checked with the transcriptions for accuracy. An independent coder will be provided with the transcripts to analyse and identify emerging themes. Tesch’s eight steps of qualitative data analysis process will be applied.

## 9. Phase II: Development of a Physical Activity Intervention Strategy

A strategy is a plan designed to achieve a particular long-term project [102]. In this study, an intervention strategy to promote PA participation in the study setting will be developed, based on the empirical findings from Phase I of the study. The development of an intervention strategy will draw on the premise of an analytical tool of the Strengths, Weaknesses, Opportunities and Threats (SWOT) and Political, Economic Growth, Socio-Cultural, Technological, Laws and Environmental (PESTLE) factors within the opportunities and threats, defining PA participation during pregnancy in Buffalo City Municipality, Eastern Cape Province. The Delphi technique will be included as an element of the validation and an adapted checklist will be used to solicit the views of key stakeholders on the developed strategy.

The SWOT analysis strategy assesses a situation to determine the “strengths”, “weaknesses”, “opportunities” and “threats” involved in a person [103]. Similarly, the PESTLE analytical tool is applied to analyse a situation and it assists in avoiding taking actions that will cause failure. The PESTLE within the SWOT context of physical activity participation during pregnancy in Buffalo City Municipality will be analysed. The Build, Overcome, Explore and Minimise (BOEM) approach will be used to develop a strategy by building from the identified strengths, overcoming the weaknesses, exploring the opportunities and minimising the threats [104].

## 10. Phase III: Validation of the Developed Intervention Strategy

The phase of validation of the developed intervention strategy will follow Phases I and II. Validation is a technique performed to determine the credibility of empirical knowledge pertaining to a scientific model of a discipline [105]. A plausible way of validating empirical knowledge could be by noting and sharing a view about what something is and how consistently it works, without formally testing the views using the research methods [105]. To facilitate validation, two key stages will be adopted. The first stage will involve the Delphi technique, while in the second stage, a checklist will be drawn up to analyse and compare findings with the initial main theme, other themes and sub-themes and the developed strategies using SWOT and PESTLE analysis strategies. The aim of this phase is to validate if the developed strategy will be used effectively to address the gaps identified during the research.

### 10.1. Delphi Technique

The Delphi technique entails a systematic, interactive process to predict future perspectives of proposed methods and strategies; it assesses the potential impact of these, if implemented [106]. The basic aim of this particular method is to utilise expert opinion on the developed strategy and implementation plans to predict their likely impact of achieving the set goals and objectives and their relevance or appropriateness [107]. This method is deemed suitable to guide strategy development in this proposed study.

We will purposively select between 5 and 10 experts, with extensive knowledge, and proven academic and scholarly background, on prenatal PA and maternal health. The findings of the study, the SWOT analysis, the PESTLE analysis strategies, the BOEM model and the subsequent developed strategies will be presented to these experts. At this point, they will be tasked to critique the developed strategy based on contextual needs and to establish the feasibility of facilitating the integration of PA for better maternal health outcomes. Subsequently, the experts’ feedback will be applied to modify the strategy in preparation for validation by key stakeholders.

### 10.2. Key Stakeholder Consultation

The key stakeholders will include managers of antenatal health clinics, antenatal midwives and pregnant women. In each selected antenatal health clinic, a health facility manager, two midwives and five pregnant women will be purposively selected to participate in the validation process. The managers from the antenatal healthcare facilities and pregnant women will constitute the population for validation of the developed PA intervention strategy. Purposive sampling will be used. The size of the sample will be based on the number of health facilities sampled. It is important to collect data from the managers of antenatal healthcare facilities because they are the agents who implement or supervise the implementation of these developed strategies in their respective healthcare facilities. Likewise, it is important to include pregnant women in the validation process because patient engagement may provide insights into possible contextual interventions that are relevant to address the individual needs of pregnant women to promote the uptake of prenatal activity. A checklist will be utilised to solicit the opinion information from key stakeholders.

Antenatal healthcare facilities’ managers, midwives and pregnant women will be invited to a workshop or meeting; the findings of the study as well as the intervention strategy will then be presented to them to discuss, deliberate on and provide their comments and opinions on feasibility, accessibility and sustainability of the proposed strategy. The stakeholders’ views will be analysed and then used to perfect the accepted strategy to promote prenatal activity for implementation in the context of Buffalo City Municipality in the Eastern Cape Province, South Africa. The strategy validation process will follow this sequence: Delphi Technique → Stakeholder Consultations → Final Strategy or Strategy for Implementation.

## 11. Ethics Approval and Consent to Participate

Ethics approval for this study has been received from the Faculty of Health Sciences Research Ethics Committee (FHREC) of the University of Fort Hare (Ref#2019=06=009=OkaforUB). In addition, permission has been obtained from the Eastern Cape Department of Health and all the selected healthcare facilities. Participants will be required to sign an informed consent form prior to data collection. Confidentiality and the anonymity of the participants and their medical, obstetric and clinical information will be maintained by not divulging their information and identity to anyone, and shall remain strictly confidential.

## 12. Discussion

Physical activity and exercise of women during pregnancy in South African women is sparsely studied. Of particular note, no known study assesses the PA participation levels, beliefs, attitudes, perceived benefits, barriers to uptake and sources of information; nor do studies explore the views of healthcare professionals concerning PA and exercise participation of women during pregnancy in the Eastern Cape Province, South Africa. Physical activity is key in promoting maternal health during the prenatal and postpartum periods, and therefore women should be encouraged to maintain an active lifestyle to avoid the health risks associated with inactiveness during pregnancy [54]. This study will make a unique contribution, in that, for the first time in the Eastern Cape Province, this study aims to assess key factors linking to PA and sedentary behaviour in pregnant women. Thus, this study will lay the groundwork for future PA intervention options during pregnancy in the Eastern Cape Province. The findings will be disseminated to women of reproductive age through meetings, workshops and social media; in addition, relevant interventions that address their needs to initiate and engage in PA during pregnancy for optimal maternal health outcomes will be shared with them.

The findings will inform obstetric care providers, policymakers and fitness professionals who provide guidance on the impact of prenatal physical activity on maternal, foetal and neonatal health outcomes.

The study will develop a context-specific PA intervention strategy for pregnant women that will address the individual experiences and needs of the study communities. In addition, the findings of this study will enable policymakers to understand the problems associated with PA and exercise participation during pregnancy among pregnant women in their setting. Such understanding is relevant to guide interventions to promote and encourage PA during pregnancy.

The results of this study may not be applicable to all pregnant populations in other provinces or regions in South Africa. In addition, because questionnaires and interviews will be used for data collection, the element of PA recall bias is possible with self-reporting. The strengths of the study will include the prospective evaluation of PA in an under-researched, poor and resource-constrained setting, and the further development of a context-specific PA strategy to guide interventions.

## 13. Conclusions

The importance of PA among pregnant women cannot be over-emphasised and, therefore, this study focuses on PA participation among pregnant women as one of the important but under-researched phenomena associated with maternal health. To be able to bridge the identified gaps, the present study is designed to assess the level of PA participation during pregnancy, and to examine the beliefs, attitudes, perceived benefits, barriers to uptake and sources of information of women during pregnancy in Buffalo City Municipality of the Eastern Cape Province, South Africa. Then, the study seeks to examine the knowledge, attitudes and practices of healthcare professionals toward PA and pregnancy in the same setting. The findings from this study may be useful in planning context-specific future PA interventions to optimise pregnancy PA, at least in this setting. Primarily, the findings will highlight the barriers to PA participation during pregnancy; such an understanding will inform context-specific interventions to address pregnant women’s needs to motivate and support them to engage in PA during pregnancy. In addition, understanding the healthcare providers’ perspectives on prenatal PA during pregnancy in the context of pregnant women in the Eastern Cape, where no empirical evidence exists, is important in guiding PA antenatal healthcare intervention. A physical activity intervention strategy will be developed to encourage and promote PA and exercise during pregnancy in the context of the Eastern Cape Province. The implementation of the developed prenatal PA strategy could improve maternal and foetal health outcomes. The dissemination of the findings of this study will be in the form of publications in peer-reviewed journals, conferences and workshops or seminars and on Twitter.

## Figures and Tables

**Figure 1 ijerph-17-06694-f001:**
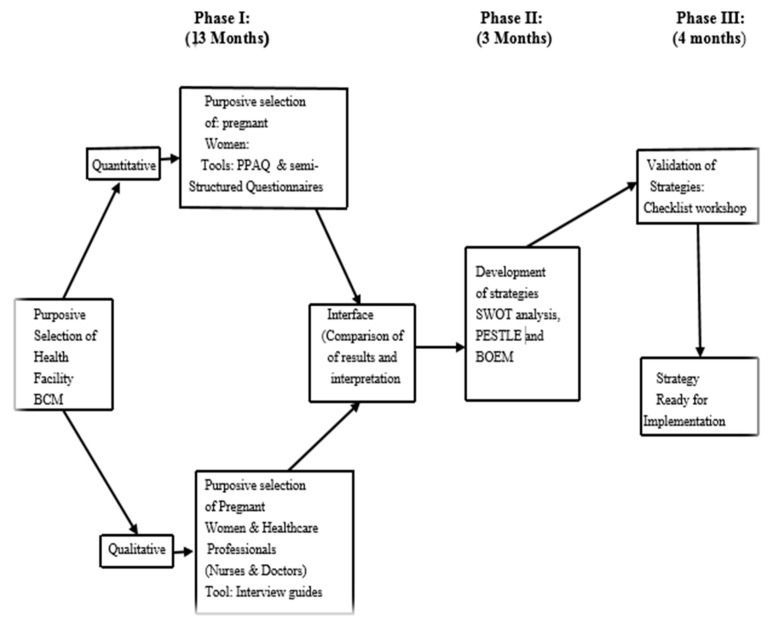
Research approach. BCM **=** Buffalo City Municipality; PPAQ = Pregnancy Physical Activity Questionnaire; SWOT = Strengths, Weaknesses, Opportunities and Threats; PESTLE = Political, Economic Growth, Socio-Cultural, Technological, Laws and Environmental; BOEM = Build, Overcome, Explore and Minimise.

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
