# Peer review of "Developing a Physical Activity Intervention Strategy for Pregnant Women in Buffalo City Municipality, South Africa: A Study Protocol"

_ijerph, 2020, doi:10.3390/ijerph17186694_

Round 1

Reviewer 1 Report

Thanks for reading this manuscript. I think it is a very important paper for the situation around the world how to get more healthy pregnant women during pregnancy which  even have impact on the life for baby and new mother. And even in the situation of great obesitas  around the world.

Author Response

Department of Nursing Science

University of Fort Hare, South Africa

August 15, 2020.

Section Managing Editor

IJERPH Editorial Office.

Mr. Mirko Kocetanovic,

RE: Manuscript ID: ijerph-887333 - Major Revisions

On behalf of my co-author, I would like to express appreciation for the thorough editorial work conducted on our protocol manuscript. The comments from all the reviewers has significantly improved the manuscript. We have tried to address Reviewer’s queries point by point as instructed and made appropriate amendments as necessary.

Once again, many thanks.

Uchenna Benedine Okafor

Reviewer 2 Report

Abstract

  • It should read Chi-square not Chi-squared
  1. Introduction
  • Page 2, first paragraph – I do not think that physical inactivity is a global burden. I think it contributes to the global burden of disease. Please revise this statement
  • Page 2, first paragraph - How does the last sentence of paragraph one fit in with the topic? The focus is on pregnant women and not young adolescents per se. Revise this sentence to focus on the pregnant population
  • Page 2, fifth paragraph – Put ‘and’ between “lack of self-confidence” and “lack of safety or fear”
  • Page 2, sixth paragraph – Put ‘and between’ “resources” and “bad weather conditions”. Remove extra period after [10, 43, 47]
  • Page 3, first paragraph – Why are women in particular at risk for sedentary lifestyles in SA?
  • It would be good to include a sentence that defines and differentiates physical activity from exercise, so it is clear to readers who may not really understand the difference. Might also be good to say something very briefly about Buffalo City Municipality
  1. Research questions
  • Page 3, first paragraph (i) – delete the word “research” between “following” and “questions”
  • Page 3, first paragraph (ii) – delete the article “the” before the word “correlates”
  • Page 3, first paragraph (v) – for consistency, use the acronym PA for physical activity
  • Page 3, last paragraph – delete the article “the” before the words “Eastern Cape”
  1. Methods/designs Aim and objectives
  • Page 5, first paragraph – let it read ‘… will focus on empirical mixed data collection from…
  • Page 5, Phase 1 – under purposive selection of health facilities, what does BCM stand for? Please provide a legend at the bottom of the figure and spell out the acronyms you have used
  1. Participant recruitment and study population
  • Page 5 – Why is the focus on women 18 years and above? Do girls under the age of 18 not get pregnant in SA?
  1. Data collection instruments
  • Page 5 – 1. Quantitative data collection – you indicate that the questionnaire is interviewer administered, but then under 7.1.2, you say the PPAQ is a self-administered instrument. If there two instruments are one and the same, please revise for consistency, so there is no confusion.
  • Page 6– 2 – How many women per focus group?
  • Page 6– 4.1 – “Two aspects of statistical analysis will be applied for data collection” replace “for” with “to”. It is Centers for ….. It is Chi-square not Chi-squared
  • Why will purposive sampling be used in the study?

Discussion

  • Page 8 – Your numbering of this section is off - .88. Please correct
  • Pages 8-9 – “The findings, if properly disseminated to women of reproductive age, pregnant women will be informed on the need to engage in PA during pregnancy for optimal health outcomes”. This sentence is unclear. It needs to be reworked

Author Contribution

  • Page 9 – “UBO conceived and wrote the first drafting”. Replace the word “drafting” with “draft”

Author Response

(The authors gave the same response as above.)

Reviewer 3 Report

This study describes the protocol for a study that will be conducted to examine physical activity patterns and beliefs in pregnant women, and to ultimately develop an intervention to enhance understanding of, and participation in physical activity during pregnancy.

The introduction has reviewed the literature extensively, however provides a lot of prior knowledge on physical activity participation, beliefs and barriers during pregnancy. Therefore it is not made clear why this study is important, beyond refining understanding to a specific population.

The aims and objectives for phase one and two are clear according to the methodology. However phase 3 (To validate the intervention strategies to promote participation of PA and exercise participation of women during pregnancy) is not clear as the research questions and methods do not indicate how the intervention will be implemented and validated.

The methodology needs expansion – the overall methodology (quantitative and qualitative) are mentioned, but there should be a clear and detailed description of the data that will be collected (only the PPAQ is described in any detail). The other data collection should be outlined clearly, the interview guides should be provided – especially since this is only a proposal and no data is available – without clear methodology this manuscript will not add much.

What outcome data was the sample size based on?

The participant section mentions that professional nurses will be included, but no information on how these will be selected, or what data will be collected from them is provided. The same applies for hospital managers, which are only mentioned at the end.

Author Response

(The authors gave the same response as above.)

Reviewer 4 Report

General:

The authors present an interesting study protocol assessing the prenatal physical activity behaviours, perceptions and experiences in women residing in the Eastern Cape, South Africa with the aim of developing an intervention to improve activity levels. I believe the proposed study is important work with appropriate scientific rigour, however, some clarification of finer details is needed. Additionally, the rationale for improving physical activity in pregnant women exists worldwide, so some specific context regarding why this is needed in this region would be helpful. At present, I find that this piece generic. I have included a number of comments that I think are important considerations for the study in general. 

Specific comments:

Abstract

  1. I would suggest that further detail regarding context i.e. why this study is specifically needed in the Eastern Cape, South Africa is required.
  2. Remove unnecessary acronyms (e.g., LMIC).
  3. Details about target recruitment size and tools, such as the PPAQ, would be useful here.

Introduction

  1. Paragraph 2: The authors have referenced a number of randomized controlled trials (RCTs)/observational studies and guideline documents within the statements in this paragraph. Some meta-analyses are referenced, however, these are not the most up to date or recent publications on this topic area. A number of comprehensive meta-analyses were published in BJSM in 2018/2019 (Davenport/Mottola) synthesising results from RCTs and other study designs on many of the outcomes listed. I would suggest that consideration of these works over choosing individual studies to reference is stronger. 
  2. In relation to the above point, some statements are referenced with ACOG guidelines for physical activity in pregnancy (e.g., GDM, DVT, low birth weight, improved sleep). I do not think citation to ACOG guidelines here is appropriate. Original sources or systematic reviews/meta-analyses pertaining to the subject should be cited only for such statements. The ACOG guidelines are roundtable expert consensus statements. 
  3. Paragraph 4: The first sentence regarding PA in women (non-pregnant?) in South Africa detracts from the focus on prenatal PA. I would suggest limiting the discussion from hereon in to pregnant women/prenatal PA.
  4. Paragraph 4: The authors cite work suggesting that 44% of pregnant women in South Africa were inactive. This would suggest 56% are physically active. This is much higher than estimates in North America (approx. 15% of pregnant women meeting guidelines; Evensen). Why is this? If this is the correct interpretation than the need for a physical activity intervention in this population is less dire than it is in other countries (although I doubt this is truly the case). The authors may want to reframe how they approach this statement for clarity. 
  5. Paragraph 6: A large barrier to physical activity in pregnant women can be childcare for other children. Will this be considered as a barrier in itself?
  6. Paragraph 7 & 8 are somewhat repetitive of each other. I think these could be condensed to key statements regarding context of why specifically, this study is needed in the Eastern Cape, South Africa. For me, these paragraphs are too generic. 

Research questions

  1. It may be planned within the questionnaires of pregnant women, but it may be worth also considering determining if women speak to their healthcare provider about PA/get recommendations AND whether women are prescribed activity restrictions by healthcare providers or choose to activity restrict. Prescription of activity restriction obviously has clear impacts on prenatal PA. 

Aims and objectives

  1. Phase 1 Aim i is quite broad. 'Characteristics' is unclear and should be specified. 
  2. Phase 2
  3. Phase 3 Aim viii. Again, this aim is broad. Some more context is needed to understand how the aim will be achieved. How exactly will the intervention be validated, through what metrics?

Methods

  • General - No description of study timelines is provided. This is important information to add.
  • Some information about the duration of the study visits and how these will occur would be useful. Are women expected to commit 1 or 5 hours to participate? This will make a significant difference in recruitment. Furthermore, will women be able to attend an antenatal appointment and participate following in the same location? Or will they need to attend a different appointment at a different location? Will childcare be provided/cost of transport be covered, will any incentives be given? At present, it is not clear how feasible this study is to achieve a target of 381 women. I would also suggest this is important methodological detail to report, as these factors will influence who is able to participate in the study (and could potentially lead to bias).

Participant recruitment and study population

  1. One of the inclusion criteria for the study is 'receiving antenatal care.' I am not familiar with the health service in South Africa, but could this potentially bias the population? One of the main outcomes of this study is considering the environment and socioeconomic context of prenatal PA, if there is a structural barrier to some women receiving antenatal care in this region, this could be a considerable limitation of the study and should be considered in future interpretation. Some women do not attend antenatal appointments because they cannot take time off work, some women do not have the funds to pay for treatment/care or do not have health insurance, some women face racial/language barriers and avoid medical appointments, etc. 
  2. Seeing as no physiological measures are being taken in this study, I am interested in the rationale for only including non-smokers. Physical activity has been used in conjunction with smoking cessation programs in pregnancy, and I find it somewhat odd that this group (who stand to benefit from lifestyle/behaviour change) would be excluded. Furthermore, smoking status can be linked to socioeconomic status, and this again may bias the recruited population. 
  3. Gestational diabetes mellitus is not a contraindication to prenatal exercise. In fact, this group can experience health benefits of PA over and beyond those of uncomplicated pregnancies. Prenatal exercise is well established as a management strategy for those diagnosed with gestational diabetes. This has not been considered a contraindication for many years across international guidelines. Please check this thoroughly. Also, provide a reference to this statement. I would also provide a defined list of contraindications so that this can be clearly identified.

Data Collection Instruments

  1. PPAQ - The PPAQ has been tested for validation across pregnant women in different countries/cultures. Do you have any concerns about its use in a South African population given its development for North American populations? This has considerable implications seeing as your data will be analyzed according to activity status. 
  2. Qualitative data - The interviews and focus group discussions for healthcare providers will be conducted 'until saturation.' This means that data analysis must occur alongside data collection. Do you have any concerns about this biasing the process? Would be it be more rigorous to identify the number of potential healthcare providers in this region and then aim to target a % of these for inclusion?
  3. Sample size - The target sample size of 381, is this pregnant women? Please be specific here. Furthermore, what is the primary outcome that this sample size calculation is based upon? More information is required.
  4. Quantitative data analysis: Your cohort will be split into inactive and active women based on PPAQ data. What expectations do you have for how many women will be included in each group? If your cohort ends up in any way similar to those of North American studies, 15% of your cohort would be considered 'active' with a majority being considered 'inactive.' How will this impact your statistical power/analysis if you end up with very unequal groups? This is especially important as your bivariate and multivariate analyses will be based on these groupings.
  5. Phase 2/3 - The authors have discussed the development and validation of the intervention, which includes consultation with healthcare providers. However, I would suggest that pregnant women are also included in this process. Patient engagement is an important area of research, and pregnant women should be able to comment on the intervention. Building this into your strategy will strengthen your design and research, as this is an increasingly important aspect of research, as well as funding opportunities.
  6. Related to funding, this work is focused on developing an intervention. It would be useful to know how this intervention will eventually be supported i.e. through grants, through the healthcare service? It is also key to note that this study is not currently funded, and is very ambitious. 

Discussion

  1. The authors mention that this work can be used to inform obstetric care providers, policymakers, fitness professionals. Although not specific to this manuscript, I would suggest that the authors develop strategies for assessing impact in each of these groups. From experience, each requires different methods of interaction/information and will be accessed through different governing bodies. Additionally, social media/local media is a useful tool for sharing research findings, so building a media strategy for reporting outcomes would be useful and cost-effective. 

Figures

  1. Figure 1: A proposed timeline of each Phase could be added. e.g. Phase 1 (6 months), Phase 2...
  2. Figure 1 and 2: Define acronyms in figure legends. 
  3. Figure 1 and 2: Potentially an issue with reading the manuscript in review format, but the quality of the figures is quite low. This could be improved. 
  4. Figure 2: This flow chart doesn't really add anything to the manuscript in its current form. This could likely be incorporated into Figure 1 in Phase 3. The long text explanations in this are not visually appealing. 

References

  1. There are citations of many different ACOG guidelines. Only the most recent should be used, seeing as this is an updated document, previous versions should no longer be used as points of reference. 
  2. As per comments re the introduction section, original sources should be cited for specific statements where possible, not reviews unless systematic or meta-analyses.
  3. I believe there are a number of conference abstracts referenced. Again, I would suggest reporting peer-reviewed full publications where possible. 

Author Response

(The authors gave the same response as above.)

Round 2

Reviewer 3 Report

Thank you for addressing my previous comments. While most of these have now been addressed, I have a few additional comments to add to this new version. Mainly – I believe restructuring of the protocol for clarity is required.

Lines 62-65 – there is something wrong with the structure of this sentence can you reword

Lines 73-77 – break up into more than one sentence; also increased maternal heart rate is no immediately clear as a benefit

Line 77 – fix “hhigher”

Line 93 – should be worryingly; check reference Most et al

Line 291 – What about disabilities or other health conditions that would not necessarily preclude PA, but would affect PA levels?

Line 319 – do you mean that the outcome is ‘adherence’ to the PA guidelines as a binary outcome? What about actual minutes of PA in different categories?

Participants – I would break the participants and inclusion criteria into two sections – 1 for the pregnant women and 1 for the healthcare providers.

Sample size – Explain exactly what outcome data was used to calculate this i.e.: what means/sds were you using and what PA data informed this sample size?

Qualitative data – again, separate pregnant women and health providers. This section is confusing – pregnant women with be interviewed in four FGDs – in what language? Using what methods? With an interview guide? Needs more detail. Then, health providers will have semi structured face to face interviews – individual or FGDs?

I believe the methods section should be separated into two sections as you have two different sample groups. You need to describe the inclusion criteria, data collection procedures (quantitative and qualitative) and sample size calculation for both of these groups separately. At the moment it is not clear and there is missing detail on certain groups.

Why do phase 2 and 3 (development and validation of the intervention) fall under the qualitative data analysis heading?

You introduce new participants and data collection methods under the Delphi method and the Stakeholder sections (which both fall under Qualitative data analysis heading)? Again – separate participants – data collection, sample size, methodology and analyses - for each different group

Alternatively – you could discuss the full methodology (including participants and data collection and analysis) separately under each unique aim.

The conclusion seems to re-state the aims. Focus more on the outcomes you expect to achieve and the impact this will have. I would not talk about implementation of the intervention here as that is outside of the scope of this study and is rather future work that may happen.

Author Response

Department of Nursing Science

University of Fort Hare, South Africa

August 22, 2020.

#Reviewer 3

IJERPH

Cc: Mr. Mirko Kocetanovic

RE: Manuscript ID: ijerph-887333 - Major Revisions

On behalf of my co-author, I would like to express appreciation for your thorough editorial work conducted on our protocol manuscript. Your further comments and suggestions have significantly improve the quality and outlook of the manuscript. We have tried to address the queries point by point as instructed and made appropriate amendments as necessary.

Once again, many thanks.

(Authors' response in the file attached)

Uchenna Benedine Okafor

Round 3

Reviewer 3 Report

NA
